# The Mechanical Fingerprint of Circulating Tumor Cells (CTCs) in Breast Cancer Patients

**DOI:** 10.3390/cancers13051119

**Published:** 2021-03-05

**Authors:** Ivonne Nel, Erik W. Morawetz, Dimitrij Tschodu, Josef A. Käs, Bahriye Aktas

**Affiliations:** 1Department of Gynecology, Medical Center, University of Leipzig, 04103 Leipzig, Germany; Bahriye.Aktas@medizin.uni-leipzig.de; 2Soft Matter Physics Division, Peter-Debye-Institute Leipzig, University of Leipzig, 04103 Leipzig, Germany; erik.morawetz@uni-leipzig.de (E.W.M.); dimitrij.tschodu@uni-leipzig.de (D.T.); jkaes@uni-leipzig.de (J.A.K.)

**Keywords:** circulating tumor cells, CTCs, optical stretcher, cell mechanics, breast cancer

## Abstract

**Simple Summary:**

Detection of circulating tumor cells (CTCs) in the blood of cancer patients is a challenging issue, since they adapt to the biochemical and physical landscape of the bloodstream. We approached the issue of CTC identification on a biophysical level. For the first time, we recorded the mechanical deformation profiles of potential CTCs, which were isolated from the blood of breast cancer patients, at the force regime of the deforming blood flow. Mechanical fingerprints of CTCs were significantly different from healthy white blood cells. We used machine learning to further evaluate the differences and identify discrimination criteria. Our results suggest that mechanical characterization of CTCs at low forces is a promising path towards CTC detection.

**Abstract:**

Circulating tumor cells (CTCs) are a potential predictive surrogate marker for disease monitoring. Due to the sparse knowledge about their phenotype and its changes during cancer progression and treatment response, CTC isolation remains challenging. Here we focused on the mechanical characterization of circulating non-hematopoietic cells from breast cancer patients to evaluate its utility for CTC detection. For proof of premise, we used healthy peripheral blood mononuclear cells (PBMCs), human MDA-MB 231 breast cancer cells and human HL-60 leukemia cells to create a CTC model system. For translational experiments CD45 negative cells—possible CTCs—were isolated from blood samples of patients with mamma carcinoma. Cells were mechanically characterized in the optical stretcher (OS). Active and passive cell mechanical data were related with physiological descriptors by a random forest (RF) classifier to identify cell type specific properties. Cancer cells were well distinguishable from PBMC in cell line tests. Analysis of clinical samples revealed that in PBMC the elliptic deformation was significantly increased compared to non-hematopoietic cells. Interestingly, non-hematopoietic cells showed significantly higher shape restoration. Based on Kelvin–Voigt modeling, the RF algorithm revealed that elliptic deformation and shape restoration were crucial parameters and that the OS discriminated non-hematopoietic cells from PBMC with an accuracy of 0.69, a sensitivity of 0.74, and specificity of 0.63. The CD45 negative cell population in the blood of breast cancer patients is mechanically distinguishable from healthy PBMC. Together with cell morphology, the mechanical fingerprint might be an appropriate tool for marker-free CTC detection.

## 1. Introduction

One reason for metastatic relapse in patients with breast cancer is hematogenous spread during early disease stages when single tumor cells detach from the primary tumor site and enter the vascular system [1]. Therefore, circulating tumor cells (CTCs) in the blood might be useful as predictive markers to monitor treatment response in the clinical setting and the risk of relapse through distant metastasis. Yet, their ability to change phenotypical, mechanical and functional properties during cancer growth and treatment leads to a variety of subpopulations that complicate the use of CTCs as a biomarker. Heterogeneity and the low concentration among blood cells make it very challenging to isolate and characterize CTCs [2]. Up-to-date no gold standard exists for the isolation of CTCs. Although controversially discussed, CellSearch is the only FDA approved assay for the detection and counting of CTCs in breast cancer patients. The procedure is based on the epithelial cell adhesion molecule (EpCAM), which is expressed on cells of epithelial origin but not on hematopoietic cells [3]. Detected cells are defined as CTCs when they are positive for the surface marker EpCAM and the intracellular cytokeratins 8, 18 and 19. Furthermore, CTCs are supposed to have a negative counterstain against the leucocyte marker CD45 and a positive nuclear DNA staining. This approach is complicated and assumes that CTCs express the same molecules as the host tissue. However, in order to leave the primary tumor site, the precursor cells of CTCs might undergo the reversible process of epithelial–mesenchymal transition (EMT), which allows them to change from epithelial to mesenchymal cells [4]. During EMT, altered polarization of the actin cytoskeleton goes along with decreased cytokeratin and increased vimentin expression. Thus, CTCs with mesenchymal or EMT-like features might be missed when using detection methods based on EpCAM [5,6]. Up to now various EpCAM-independent approaches like density gradient centrifugation [7], microfiltration dependent on size and deformability [8], other 2D- and 3D membrane microfilters [9,10], microfluidic approaches using bioelectric properties [11] and several CTC chambers, -channels and -chips [12,13,14] commonly based on distinct cellular and biophysical differences between CTCs and blood cells are available [15,16,17,18]. A promising method is the so-called negative CTC selection by depletion of hematopoietic cells using magnetically labeled antibodies against CD45 to enrich non-hematopoietic cells in the remaining cell suspension [19,20]. These non-hematopoietic cells include possible CTC candidates with not only epithelial but also mesenchymal- or stem-cell-like characteristics. Morphological criteria define cancer cells as cells with an enlarged nucleus-to-cytoplasm ratio and a large cell size [21]. These cytological properties are associated with altered mechanical characteristics such as dynamic modification of the cytoskeletal stiffness [22]. Cancer cells exhibit a lower mechanical resistance with respect to healthy tissue [23,24] and hence might be privileged to migrate into distant organs [25]. Additionally, surface charge and electrical properties were reported to be affected by altered cellular stiffness [26]. The cytoskeleton of eukaryotic cells is highly adaptive—even without changes in gene expression regulated by its accessory proteins—causing variability in cell behavior without phenotypical changes [27]. Due to phenotypical, mechanical and functional changes during cancer progression and treatment, isolation of CTC subpopulations remains challenging and cannot be guided by the phenotype alone; cell mechanics have to be considered equally.

The purpose of the present study was to characterize the mechanics of non-hematopoietic cells from breast cancer patients to evaluate the utility of material parameters for CTC separation from blood. It has been shown that separation from white blood cells based on mechanical properties is possible for the leukemia cell line Jurkat and for the breast and prostate cancer cell lines MDA-MB-231 and LNCaP C4-2, [22,24]. For the first time, we investigated the cell deformation behavior concerning the active and passive mechanical resistance of CD45 negative cells from the blood of breast cancer and present our findings as a proof of premise of mechanical phenotyping. The OS allows quantitative measurement of mechanical resistance of cells in suspension. By a Gaussian dual beam trap cells are deformed non-invasively and the relative deformation is measured in a step stress creep experiment [24,28,29,30,31,32,33,34]. The OS works with a low pulling stress at the order of one Pa, stretching the cells by a few percent over the duration of several seconds, similar to the deforming blood flow [35,36]. Cells exhibit passive viscoelastic and active mechanical properties. Therefore, rheological models, such as our extended Kelvin–Voigt model, which combines a linear elastic spring and linear viscous dash pot with an active linear contractility term, can be applied to analyze the cell mechanics of the measured cells [37]. Furthermore, a random forest machine learning algorithm, which is a cascade of individual decision trees, has proven to be robustly applicable as a test of the prediction power to detect non-hematopoietic cells [38]. We were able to show that defining mechanical fingerprints of non-hematopoietic cells might be a promising approach to detect CTCs in patients’ blood. By choosing no other identifier for possible breast cancer derived CTCs than the absence of CD45, we could characterize these rare cells on the most basic level. A more focused study of non-hematopoietic cells carrying different markers that have been reported to be associated with CTCs would complement the prediction power even further.

## 2. Materials and Methods

### 2.1. Study Population, Blood Samples and Informed Consent

Patients with mamma carcinoma who received anticancer treatment in the University of Leipzig Medical Center were consecutively included in this study after agreeing and signing a written informed consent in accordance with the requirements of our institution’s board of ethics (internal reference number: No. 216/18-ek). Patient demographics are described in Table 1.

### 2.2. Cell Culture

Stably transfected GFP-expressing MDAMB 231 cells were obtained from Cell Biolabs, Inc. (San Diego, CA, USA) and maintained under standard conditions at 37  °C in a 95% air and 5% CO_2_ atmosphere [39,40]. Cells were cultured in DMEM containing 4.5 g/L glucose, l-glutamine (Cat.No. FG 0435, Biochrom, Cambridge, United Kingdom) supplemented with 10% fetal bovine serum (Cat.No. S 0615, Biochrom) and 100 U/mL penicillin/streptomycin. HL-60 cells were obtained from ATCC (Manassas, VA, USA) and maintained under standard conditions at 37  °C in a 95% air and 5% CO_2_ atmosphere suspended in DMEM containing 4.5 g/L glucose, l-glutamine (Cat.No. FG 0435, Biochrom) supplemented with 20% fetal bovine serum (Cat. No. S 0615, Biochrom) and 100 U/mL penicillin/streptomycin.

### 2.3. Sample Preparation and CTC Enrichment

For a paradigmatic test system, healthy peripheral blood mononuclear cells (PBMCs) were isolated from whole blood by buoyant density gradient centrifugation (1600× *g*, 20 °C, 20 min). Epithelial MDA-MB 231 breast cancer cells were detached using 0.025% Trypsin/EDTA (PAA) and resuspended in culture medium. HL-60 leukemia cells are cultured in suspension and were corrected for the right concentration by centrifugation and resuspension. MDA-MB-231 and HL-60 cells were mixed with healthy PBMC, respectively, to mimic CTCs in the OS measurement.

For translational experiments, blood samples were collected from breast cancer patients one day before surgery and processed within 12 h after collection. Briefly, 10 mL peripheral venous blood were diluted with 10 mL PBS and carefully layered into a tube containing 16 mL Ficoll–Paque (GE-Healthcare, Buckinghamshire, United Kingdom) below a porous barrier. After buoyant density gradient centrifugation (1600× *g*, 20 °C, 20 min) the interphase consisting of PBMC and CTCs was removed and washed. The remaining red blood cells were removed using magnetic particles coated with antibodies against human glycophorin a (CD235a MicroBeads, 130-050-501, Miltenyi Biotec, Bergisch Gladbach, Germany). The cell suspension was then incubated with microbeads directed against human CD45 to deplete hematopoietic cells according to the manufacturer’s instructions (CD45 MicroBeads, 130-045-801, Miltenyi Biotec, Bergisch Gladbach, Germany; Figure 1).

### 2.4. Cell Rheological Measurements

Details of the Optical Stretcher are described elsewhere [26,41]. For cell line tests, we measured cells at 875 mW laser power for 5 s at 37 °C cell temperature during the measurements. A stretching time period of 5 s, during which the cell is exposed to a step stress, is sufficient to capture the cell’s entire mechanical fingerprint including active contractions. After the step stress, laser power was set back to trapping power that just holds the cell, and the relaxation of the cell from being stretched was observed for 2 s. Since differences in the relative deformation of our clinical samples between CTCs and PBMC were not as clear, we decided to increase the measurement time to 10 s and added a set of three different step stresses generated by increasing laser powers. During the measurement of a sample, for each individual single cell measured, the laser power was set in random order to either 400, 800 or 1200 mW. Observation time for the relaxation was again 2 s. For every single cell, the deformation and relaxation behavior were recorded using phase contrast microscopy simultaneously obtaining additional data such as cell size and brightness of the cell body. An edge detection algorithm tracked the short and the long axis of the cells during their deformation. From the relative change of the long axis the relative deformation is calculated. The elliptic deformation is then defined as the relation between the long axis and short axis. Plotted over time, these rheological parameters describe the deformation of a cell during the step stress and relaxation phase.

### 2.5. Kelvin–Voigt Fitting

To numerically describe the deformation behavior, we fit a viscoelastic Kelvin–Voigt model to the deformation curves [34,42]. In its original form, it consists of a spring in parallel to a dashpot. We extended this model by introducing an active contraction of the cell in response to the step stress resulting in a linear decrease of the step stress with time, since the contraction counteracts the optical stretching force. We refer to this additional contractility parameter as activity in Pa/s [34]. These models were fitted to the relative and elliptic deformation curves over the entire duration and over the first 2 s of the step stress period. For examples of Kelvin–Voigt fitting see Appendix A, Figure A2. The obtained passive and active rheological parameters for the tested cells and the goodness of the model fit were fed into the data matrix of the random forest (RF) algorithm to evaluate whether these parameters distinguish CTCs from blood cells.

### 2.6. Machine Learning

CTC candidates and PBMC were classified using the RF machine learning algorithm [43], which consists of various individual decision trees, whereby each tree makes a single prediction after splitting the data according to a purity measure. The RF is particularly suitable for our study, since this algorithm does not assume any underlying distribution of the data. As an ensemble of decision trees, it presents a robust algorithm meaning that outliers have minor influence on predictions and examination of decisive parameters is possible. For a detailed explanation of all features and the prediction procedure, please see Appendix B.

### 2.7. Statistical Analysis

All significance tests presented in this study are two sample Kolmogorov–Smirnov tests, as distributions were typically not normal. They were performed using Matlab 2019b.

## 3. Results

### 3.1. Optical Stretching Reveals Significant Differences in the Resistance between White Blood Cells, MDA-MB 231 and HL-60

In proof of premise experiments, we tested whether discrimination of malignant cells and white blood cells using the OS is possible, akin to other mechanical techniques [22,23]. In a first step, we investigated mechanical profiles of PBMC samples from healthy donors (*n* = 3) compared to PBMC from breast cancer patients (*n* = 2). The mechanical properties were without significant differences (Figure 2A). Therefore, data obtained from PBMC measurements were pooled to serve as a reference for further analysis (*n* = 5). Subsequently, we measured the mechanical characteristics of epithelial breast cancer cells from the highly invasive cell line MDA-MB 231, which represents a mesenchymal-like phenotype, and HL-60 leukemia cells, which are naturally habitant in blood. All cell populations, PBMC, MDA-MB 231 cells and HL-60 cells behaved mechanically differently in the OS. Comparing the three deformation patterns, we were able to establish significantly disparate mechanical profiles (*p* < 0.001; Figure 2B). In comparison to MDA-MB 231 cells, PBMC were much softer and showed more than a twofold elevated relative deformation (median relative deformation MDA-MB 231 = 0.012, median relative deformation PBMC = 0.028). HL-60 cells showed a median relative deformation of 0.023 entailing that they were softer than breast cancer cells but stiffer than PBMC. We quantitatively described the cell deformation behavior using our extended Kelvin–Voigt model (Appendix A, Figure A2) and added optical and morphologic descriptors obtained from phase contrast imaging such as cell radius, brightness of the cell and initial shape to our analysis (for complete description of the parameters used, please follow the link in Appendix B). The resulting data matrix consisted of 5284 cells with 76 parameters each and was analyzed by a RF algorithm to distinguish whether individually measured cells were PBMC, HL-60 or MDA-MB 231 cells. The RF distinguished these cell types with an accuracy of 0.93, a sensitivity of 0.86 and a specificity of 0.96. Excluding the morphological and optical parameters cell radius, cell area and relative cell brightness, which were decisive for the prediction, accuracy decreased to 0.78, sensitivity to 0.66 and specificity to 0.86. The following parameters were important for the classification: (1) Degree of rotation during the experiment: some cells, when irregularly shaped, rotate around one or multiple axes during the stretch. (2) Elasticity: Young’s modulus as determined by the Kelvin–Voigt fits. (3) Shape restoration: cells tend to contract towards their initial shape after being elongated in the optical stretcher, which can be quantified by subtracting the elongation after relaxation from the elongation after stretching. (4) Fit errors: when cells exhibit a very clear viscoelastic deformation behavior, the Kelvin–Voigt fits do not differ much from the actual deformation curve—when the deformation is noisy, or cannot be well described by the models, the error gets larger.

### 3.2. Elliptic Deformation and Shape Restoration Discriminate Blood Cells and CTC Candidates from Mamma Carcinoma

After demonstrating that the OS permits cell-mechanics-based discrimination of cancer cells from white blood cells, clinical samples from patients with breast cancer (*n* = 12) were investigated. Density gradient centrifugation followed by depletion of erythrocytes and leucocytes using microbeads against glycophorin a and CD45 were performed as described in the methods section. Subsequently, the remaining non-hematopoietic cells, which are potential CTCs, were measured in the OS. The mean proportion of CD45 negative, non-hematopoietic cells in relation to the CD45 positive, hematopoietic population was 3.26% in samples from breast cancer patients, and 0.25% in samples from healthy donors (Appendix A, Table A1). In total, we analyzed 3641 non-hematopoietic cells derived from 12 patients with breast cancer. Measurements were carried out using three different laser powers P_1_ = 400 mW, P_2_ = 800 mW and P_3_ = 1200 mW, resulting in three different step stresses (estimates of the step stresses: σ_1_ = 0.38 Pa, σ_2_ = 0.76 Pa and σ_3_ = 1.14 Pa, corresponding estimates of the peak stretching force: F_1_ = 80 pN, F_2_ = 160 pN and F_3_ = 240 pN) [26]. Testing cells at three different laser powers covers a broader force regime, which may trigger different responses from cells. In total, 2541 CD45 positive PBMC from three healthy donors and two patients with breast cancer were measured under the same conditions as a control. For each laser power (400, 800 and 1200 mW) the combined deformation curves of non-hematopoietic CTC-candidates from mamma carcinomas compared to PBMC are shown in Figure 3A–C. The relative deformation behavior of the different cells at the three stretching powers appeared to be quite similar. Nevertheless, comparison of the cell type specific elliptic deformation draws another picture. The relative deformation takes only the process of elongation into account. The Poisson effect that describes the shortening or expansion of a material in directions perpendicular to the direction of stretching illustrates that elongation alone falls short as a descriptor of cell resistance. Elliptic deformation, however, describes the change in cell shape in the OS more precisely. The ellipticity of CTC candidates and PBMC differed significantly at each laser power. For PBMC the elliptic deformation was increased by a factor of 2 (*p* (P = 400 mW) = 0.001, *p* (P = 800 mW) < 0.001 and *p* (P = 1200 mW) < 0.001), Figure 3E–F. In addition, we analyzed the mechanosensitive active response of a cell triggered by external forces induced by the laser trap. The active Kelvin–Voigt model can be applied for active, linearly increasing counter forces caused by a radially contracting cell in suspension and was used to estimate the contraction force of a cell opposing optical stretching [34]. Active contractions were equally observed in all cell types at all force regimes. Median activity values of cells were 0.079 Pa/s for CTC candidates from patients with breast cancer and 0.074 Pa/s for PBMC. The ratio of active to non-active cells in all samples was 0.6 ± 0.05 (SD) and did not differ from this range between all measured cell types. A cell was considered active when the activity parameter given by the active Kelvin–Voigt model exceeded a threshold of 0.001 Pa/s. Values below that threshold were considered non-active, as lower values vanish in the random noise of the fit. After the end of being stretched by the laser-induced step stress, cells were tending to restore their original shape during relaxation (Figure 4). An adequate measure of shape restoration was the difference between the elongation at 1.5 s after the end of optical stretching compared to the elongation at the end of the step stress. Interestingly, we revealed that PBMC showed significantly decreased shape restoration, i.e., a more viscous, dissipative behavior, compared to CTC candidates at 400 mW and 1200 mW (*p* < 0.001, Figure 5).

The number of non-hematopoietic cells was 13-fold increased in blood samples of breast cancer patients compared to healthy donors, indicating that they contain a significant portion of possible CTCs. Yet, to ensure that the analyzed population of non-hematopoietic CTC candidates represents actual CTCs, we performed immunofluorescence staining using antibodies against pan-cytokeratin that react with all types of epithelia (Appendix A, Figure A3).

### 3.3. CTC Candidates Can Be Distinguished from Blood Cells by Machine Learning

In analogy to our test with malignant cell lines, we applied our machine learning algorithm to predict cell type origin—hematopoietic or mamma carcinoma—based on OS measurements. Using a data set consisting of 3641 non-hematopoietic CD45 negative cells from breast cancer patients and 2541 PBMC, we classified these cells into CTCs and PBMC by their mechanical fingerprint using the RF algorithm [38]. The input chosen for the data matrix is described in the Materials and Methods section and as already mentioned included morphological and optical parameters such as cell size and brightness and a detailed set of numerical data describing the cells’ mechanical resistance using viscoelastic modeling. Separate analysis was performed for each laser power and for the complete set of data, pooling all laser powers. Based on the extended Kelvin–Voigt model, the RF algorithm revealed that the OS was able to detect CTCs with an average accuracy of 0.66, a sensitivity of 0.74 and a specificity of 0.55 (Table 2). For all three laser powers (400, 800 and 1200 mW), cells were identified by the RF with a moderate accuracy, albeit the most distinguishing parameters varied conditionally on the laser power. For all three laser powers, the most important determining feature was the cell radius, followed by the elliptic deformation at the end of the stretching phase. For the cells’ deformation behavior, elliptic deformation is the superior dividing feature with respect to the relative deformation, which was already shown in Figure 3. When data from all three laser powers were pooled before applying the RF algorithm, accuracy was 0.66, sensitivity was 0.77 and specificity was 0.53. Interestingly, when shape restoration/relaxation was included in the machine learning analysis, it appeared to be the second most decisive parameter at 800 and 1200 mW, next to cell radius. At the highest laser power, accuracy, sensitivity and specificity of the prediction were 0.69, 0.74 and 0.63, respectively, which can be considered the best result (Table 2). The impact of morphological and mechanical effects is summarized in Table 3, which shows prediction performances of the step-wise data input, starting with the morphological parameters cell area and cell radius only. To these, we progressively added the following mechanical features: relative deformation, elliptical deformation and shape restoration, and subsequently computed the prediction results for the pooled laser powers. No other parameters were considered at this point. We revealed better accuracy, sensitivity and specificity of the prediction when elliptical deformation and shape restoration parameters were included.

## 4. Discussion

Using the OS, we were able to reveal distinct mechanical fingerprints for hematopoietic cells and non-hematopoietic cells from breast cancer patients. We established that optical deformability could be used to distinguish PBMC from the cancerous cell lines MDA-MB 231 and HL-60. In these model systems, cell types could be distinguished with a sensitivity of 0.86 and a specificity of 0.96 based on relative deformation in conjunction with a subsequent evaluation relying on Kelvin–Voigt models at two different time scales and the RF machine learning algorithm. Interestingly, PBMC turned out to be twice as soft as the highly metastatic MDA-MB 231 cells, which in turn were distinctively softer than their healthy epithelial precursors. It has been shown before, that the optical stretcher is able to resolve those mechanical changes [33]. That cancer cells, once the disease was initiated, altered their cytoskeleton and mechanical response has been reported for clinical samples and different carcinomas [44,45,46]. Breast cancer cells specifically were softer than healthy epithelial cells, while mechanical response was more heterogeneous in general [23,24]. These changes arose from the remodeling of the cytoskeleton after cancer associated signaling pathways were activated and provided the basis for our selection approach. Further, HL-60, a leukemia cell line, appeared to be mechanically closer to PBMC. Still, the OS was able to discriminate the different cell types accurately. Hence, we can trust differences that we found between CD45 negative cells derived from breast cancer patients and healthy CD45 positive PBMC. While MDA-MB 231 and HL-60 cells significantly differed from PBMC in terms of their relative deformation, CD45 negative CTC candidates needed higher optical stretching forces for better resolution. CTC candidates, however, showed significantly different behavior from PBMC in terms of the elliptic deformation at all laser powers.

Our analysis of clinical samples from breast cancer patients provided proof of premise that identification of CTCs based on the mechanical fingerprint is principally possible. There was no way to ensure that all non-hematopoietic CTC candidates measured in the OS were of cancerous origin. However, the ratio of non-hematopoietic to hematopoietic cells separated by our protocol was 14-fold higher for breast-cancer patients compared to healthy donors. Immunofluorescence staining against cytokeratins 7, 8, 18 and 19 confirmed a subpopulation of epithelial cells in the blood of breast cancer patients. Hence, the significant descriptors that we revealed can be applied for reproducible identification of CTCs. The relative deformation significantly differed between non-hematopoietic CTC candidates and white blood cells only at the highest laser power. This indicates that the cellular softness of non-hematopoietic cells is relatively close to PBMC. Non-hematopoietic cells from breast cancer patients appeared to mimic healthy blood cells, as they are very similar to blood in their mechanical resistance. It makes sense that cancer cells, which make up a large part of the non-hematopoietic population in patients with breast cancer, would adapt to properties of PBMC during their travel along the blood stream, as cancer cells are already in a state of mechanical alteration. This closeness in the mechanical fingerprints between CTC candidates and PBMC might explain why existing CTC isolation approaches based on mechanical differences such as filters or high throughput microfluidic deformation of cells utilizing large mechanical deformations are still in need of improvement. A more subtle approach that can resolve smaller mechanical differences is required.

The elliptic deformation allowed the detection of CTC candidates among PBMCs. The elliptic deformation is a more refined description of the mechanical resistance of a cell since it considers the Poisson effect [32]. Not only pure elongation, but also plasticity is taken into account, which appears to be a more sensitive measure. Interestingly, the ability to restore shape after deformation was significantly higher in CTC candidates derived from patients with mamma carcinomas compared to PBMC at 400 mW and 1200 mW. The combination of OS derived data and subsequent machine learning allowed for a label free discrimination of cell types with a sensitivity of 0.74 and a specificity of 0.63, meaning CTC candidates were easier to identify than PBMC. For all laser powers, the most predictive factor was the cell radius, which goes along with the classic morphological features of white blood cells compared to tumor cells. Yet, morphological parameters alone were not sufficient for a prediction that is notably better than random guessing (accuracy of a random classifier = 0.5 for two classes and accuracy of the prediction with morphological parameters only = 0.55). Proper prediction was only achieved with the use of rheological properties. It also became clear that the prediction considerably increased with increasing laser power. At 1200 mW, shape restoration was the top distinguishing feature after the cell radius. This arises from the better resolution that comes with higher forces, ergo larger deformations.

The mechanical differences between CTC candidates and PBMC appeared to be small since cancer cells tend to adapt to the blood stream; yet it was possible to resolve them using the OS. In addition, our effort in mechanical CTC characterization relies on only one, very low-level, necessary but not sufficient, initial criterion: CTC candidates were not expressing CD45 on their surfaces. We cannot tell whether all analyzed cells were of cancerous origin, yet we can be sure that we did not exclude any candidates by prior labeling and sorting, except for dual positive, CD45 expressing CTCs [20,31,35]. We are comparing a hematopoietic cell population that does not contain CTCs (CD45+) to a subpopulation that does probably contain the major proportion of CTCs (CD45−). Our chosen approach will most likely benefit from further biochemical characterization of the separated cell populations in future projects. Still, our results highlight two key differences: CTC candidates were larger and mechanically more resistant in the blood stream. Recent microfluidic results [37,38] have demonstrated that larger and more resistant cells are pressed towards the vascular walls or into smaller blood vessels by the blood stream. To which extent these properties of non-hematopoietic CTC candidates, that might foster contact with vessel walls, promote vascular extravasation or even metastatic spread needs to be further investigated. Here, we presented a proof of premise study using clinical samples from breast cancer patients. Surely, further clinical investigations are needed to strengthen our CTC detection approach. While the prediction scores require improvement, it remains noteworthy that significant cellular discrimination is possible using this very basic experimental setup. Likely, more high throughput mechanical techniques may be helpful [41,47].

## 5. Conclusions

Here we described which cell mechanical properties might be used as a potential tool to discriminate tumor cells from blood cells in the clinical setting. Although CTCs were difficult to track, their mechanical fingerprint allowed us to identify non-hematopoietic cells in the blood of patients with mamma carcinoma. The OS enabled us to perform contact free, sensitive mechanical testing of individual cells and, hence, to discover distinct differences between active and passive mechanical properties among cell types. Whereas previous studies postulated that the relative deformation of a cell might be predictive for cancerous characteristics, we found that mechanical analysis improved the characterization of single cells of possibly cancerous origin [33,44,45,46]. Particularly elliptic deformation and shape restoration at high force regimes allowed discrimination of CTCs. Although the OS device is not suitable for high throughput it might be useful to evaluate mechanical separation parameters and ultimately plays a role in the development of label-free isolation approaches including filtration by size or separation by cellular compressibility.

Therefore, we conclude that together with cell morphology, mechanical deformation patterns might be an appropriate tool for marker-free CTC detection in the peripheral blood of patients with breast cancer. The promising combination of OS and random forest analyses might be enhanced with diagnostic cellular markers to potentially improve future treatment decisions.

## Figures and Tables

**Figure 1 cancers-13-01119-f001:**
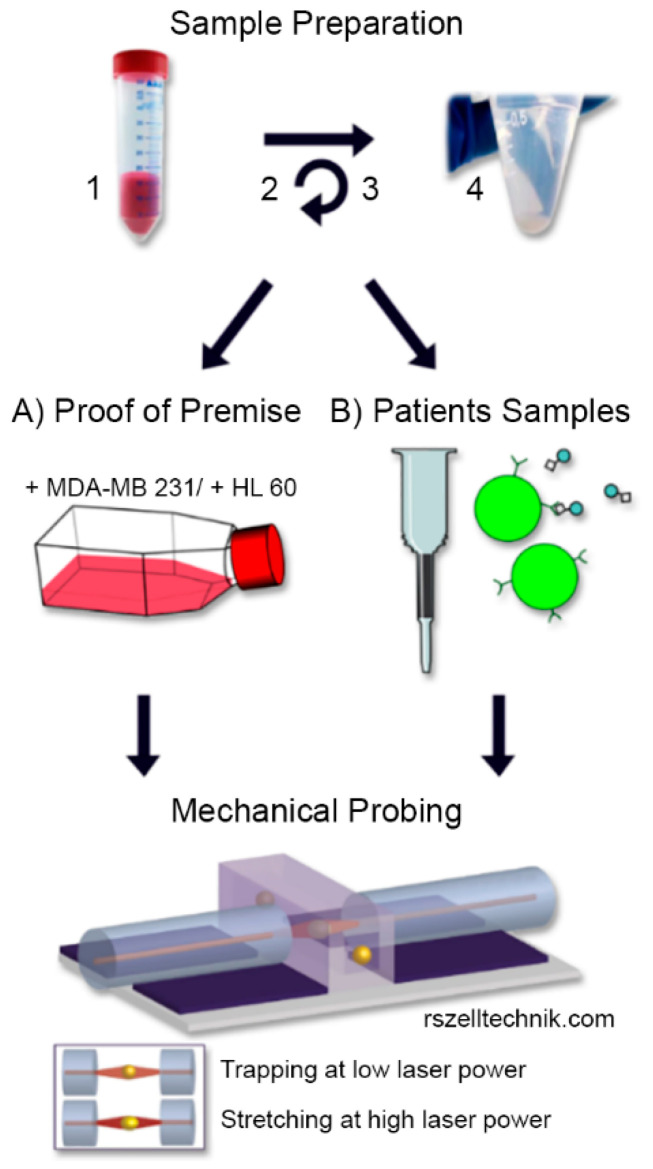
Sample preparation and measurement: 1. Whole blood sample diluted with PBS; 2. density gradient centrifugation; 3. depletion of red blood cells using microbeads against glycophorin a and 4. resulting PBMC were resuspended and (**A**) spiked with MDA-MB 231 breast cancer cells and HL-60 leukemia cells for proof of premise experiments to mimic CTC samples or (**B**) incubated with microbeads against CD45 to deplete hematopoietic cells and thus enrich possible CTC candidates from clinical samples. Spiked respective remaining cell suspensions were applied to the optical stretcher and rheological parameters were measured.

**Figure 2 cancers-13-01119-f002:**
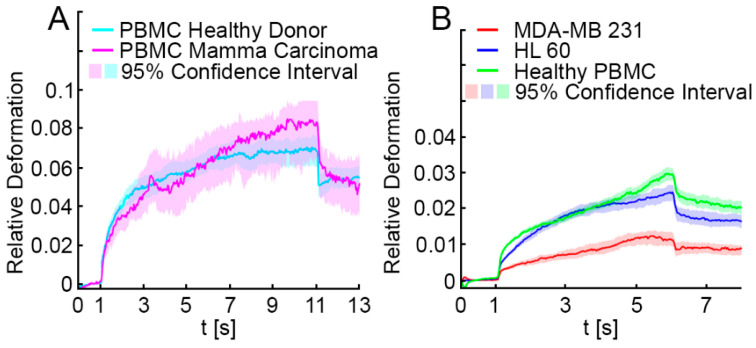
Relative deformation curves of blood cells and MDA-MB 231 and HL-60. (**A**) PBMC from healthy donors (cyan line, *n* = 3) and from patients with breast cancer (magenta line, *n* = 2) exhibited the same deformation behavior, here shown in a representative example at 1200 mW stretching power. Data obtained from PBMC measurements were pooled to serve as a reference for further analysis (*n* = 5). Relative deformations and elliptic deformations of the samples were not distinguishable (for elliptic deformations see Appendix A, Figure A1). (**B**) We analyzed the relative deformation of PBMC samples (green line). We then measured the relative deformation of breast cancer cells from the highly invasive breast cancer cell line MDA-MB 231 (red line) and the leukemia cell line HL-60 (blue line). The deformation patterns of the three cell types were significantly different (*p* < 0.001) at 875 mW. In comparison to MDA-MB 231 cells, PBMC were much softer and showed a more than twofold elevated relative deformation (median relative deformation MDA-MB 231 = 0.012, median relative deformation PBMC = 0.028). HL-60 was softer than MDA-MB 231, but stiffer than PBMC (median relative deformation HL-60 = 0.023). Elliptic deformation showed the same trend and statistically significant differences (*p* < 0.001, Appendix A, Figure A1).

**Figure 3 cancers-13-01119-f003:**
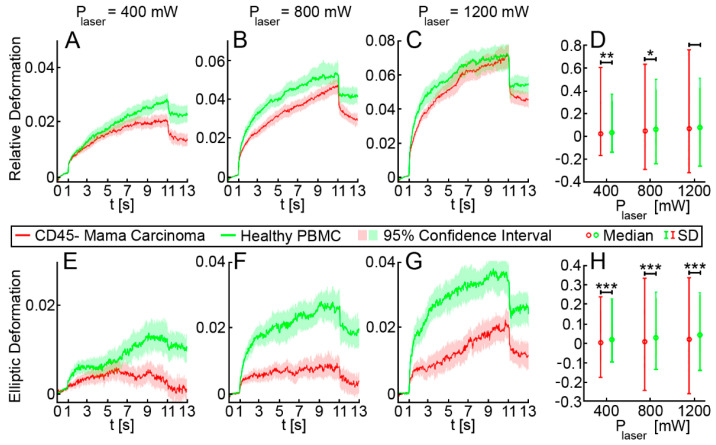
Relative and elliptic deformation curves of blood cells compared to non-hematopoietic CTC candidates from mamma carcinoma patients. In total, we analyzed 2541 CD45 positive PBMC from 3 healthy donors and 2 patients with breast cancer (green lines) and 3641 non-hematopoietic cells derived from 12 patients with breast cancer (red line). Measurements were carried out using three different laser powers P1 = 400 mW, P2 = 800 mW and P3 = 1200 mW, resulting in three different step stresses. (**A**–**C**) show the pooled relative deformation curves of possible CTC candidates from mamma carcinoma compared to PBMC at each laser power. The deformation behavior of the two cell populations appeared to be similar, and only at P = 400 mW and P = 800 mW the deformation curves differed significantly at the end of the stretching phase (*p* = 0.001), as shown in (**D**). (**E**–**G**) show the elliptic deformation of the two cell populations at each laser power. The ellipticity of CTC candidates and PBMC differed significantly at each laser power indicating that in PBMC the elliptic deformation was increased by the factor 2 (*p* (P = 400 mW) < 0.001, *p* (P = 800 mW) < 0.001 and *p* (P = 1200 mW) < 0.001). (H) Elliptic deformation of CTC candidates derived from patients with mamma carcinoma was significantly lower at all laser powers compared to CD45 positive PBMC. Asterisks indicate significance of the Kolmogorov-Smirnov-test (*—*p* < 0.05; **—*p* < 0.01; ***—*p* < 0.001).

**Figure 4 cancers-13-01119-f004:**
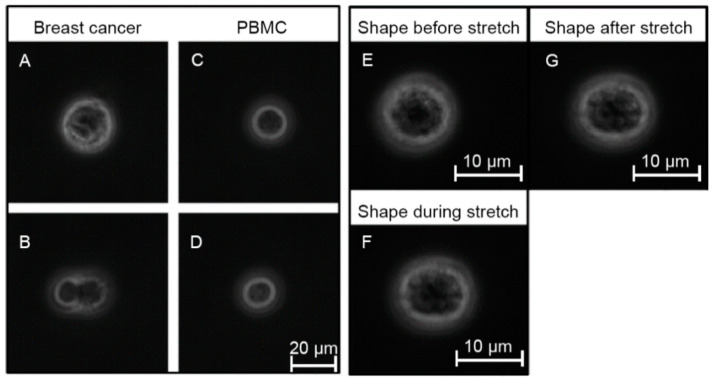
Representative phase contrast images obtained during mechanical probing using the optical stretcher. (**A**) Possible CTC candidate and (**B**) cell cluster—both detected in the CD45 depleted cell suspension from a breast cancer patient. (**C**,**D**) show PBMC after density gradient centrifugation, (**E**–**G**) show the shape of a breast cancer derived CTC candidate with particularly low mechanical resistance before, during and after optical stretching demonstrating that mechanical changes are very small.

**Figure 5 cancers-13-01119-f005:**
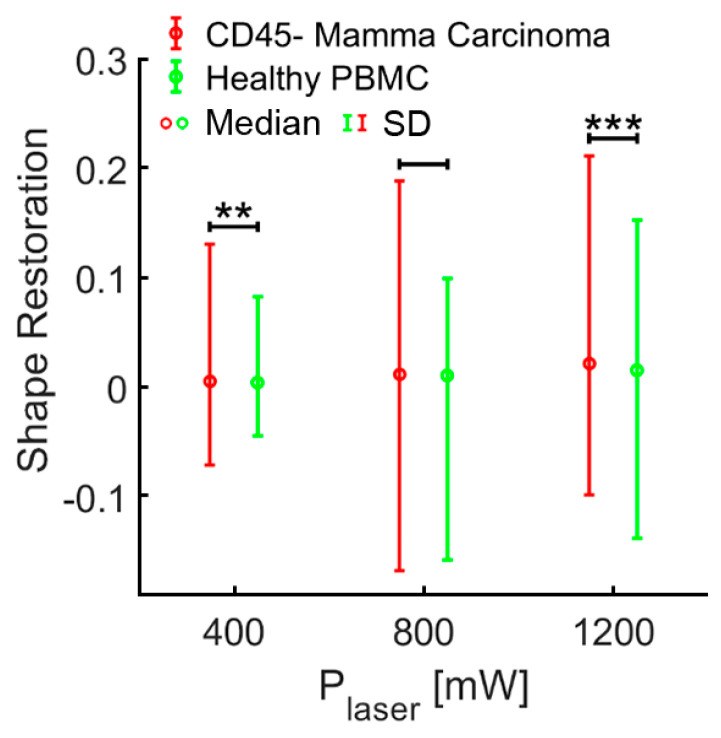
Shape restoration of cells in the optical stretcher. After being exposed to the laser-induced step stress, cells were tending to restore their original shape. An adequate shape restoration parameter was the difference between the elongation at the end of the step stress and the elongation at 1.5 s after the step stress. At 400 mW and 1200 mW, PBMC showed significantly lower shape restoration compared to CTC candidates from mamma carcinoma (400 mW: *p* < 0.01, 1200 mW: *p* < 0.001). Asterisks indicate significance of the Kolmogorov-Smirnov-test (**—*p* < 0.01; ***—*p* < 0.001).

**Table 1 cancers-13-01119-t001:** Clinico-pathological characteristics. Patients were included after agreeing and signing a written informed consent.

Characteristic	Mamma Carcinoma
	(*n* = 14) *
Age (years)	26–85
Median	63
Histology	
ductal	10
lobular	2
mixed	2
Stage	
T0	1
Tis	3
T1	7
T2	3
Lymphnode	
N0	10
N1	2
Nx	2
Grade	
G1	5
G2	4
G3	5
Subtype	
luminal A	4
luminal B	4
HER2 enriched	1
triple negative	3
DCIS	2
Setting	
primary	12
recurrent	2
Therapy	
neoadjuvant CT	3
neoadjuvant endocrine	3
adjuvant CT	1
adjuvant endocrine	9
adjuvant RT	1
adjuvant Trastuzumab	1
none	1

* For proof of premise experiments, we analyzed peripheral blood mononuclear cells (PBMCs) obtained from 3 healthy donors compared to 2 breast cancer patients. For translational experiments, we used blood samples from patients with mamma carcinoma (*n* = 12). Hence, the experimental design results in the total number of 14 patients with mamma carcinomas (*n* = 14).

**Table 2 cancers-13-01119-t002:** Prediction performance of the random forest machine learning algorithm. The algorithm was applied to our optical stretcher data set of 3641 non-hematopoietic cells from breast cancer patients and 2541 PBMC. Cells were tested in step stress experiments at various laser powers, and physiological parameters and cellular deformation were recorded. Kelvin–Voigt modeling was applied to the data to derive a number of active and passive rheological parameters. The prediction power slightly increased from 400 mW over 800 mW to 1200 mW. When data from all three laser powers were pooled before applying the RF algorithm, accuracy was 0.65, sensitivity and specificity were 0.73 and 0.56, respectively. Interestingly, when shape restoration/relaxation was included, accuracy, sensitivity and specificity increased to 0.69, 0.74 and 0.63, respectively, at the highest laser power.

Laser Power (mW)	Excluding Shape Restoration			Including Shape Restoration		
	Accuracy	Sensitivity	Specificity	Accuracy	Sensitivity	Specificity
400	0.63	0.72	0.52	0.64	0.73	0.52
800	0.66	0.76	0.55	0.65	0.76	0.51
1200	0.66	0.72	0.60	0.69	0.74	0.63
averaged	0.65	0.73	0.56	0.66	0.74	0.55
pooled	0.65	0.72	0.56	0.66	0.77	0.53

**Table 3 cancers-13-01119-t003:** Progressive input of parameters of pooled data. To distinguish morphological from mechanical effects, we first tested the prediction power of the morphological parameters cell area and cell radius only, and then progressively added the mechanical features relative deformation, elliptical deformation, and shape restoration. The prediction results were computed for the pooled data and revealed superior accuracy, sensitivity and specificity when specific mechanical parameters were included. The values at the last step do not match the corresponding values in Table 2, since here we added only the first three most important mechanical parameters and the prediction in Table 2 contained numerous additional parameters

Stepwise Data Input	Accuracy	Sensitivity	Specificity
Cell area, cell radius	0.55	0.61	0.48
+Relative deformation	0.57	0.61	0.53
+Elliptic deformation	0.59	0.65	0.52
+Shape restoration	0.62	0.70	0.55

## Data Availability

Our data, lists of all parameters, tables of all permutation importances, and code used for the machine learning analysis can be found at https://github.com/DiTscho/OS.

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
