# Peer review of "The Mechanical Fingerprint of Circulating Tumor Cells (CTCs) in Breast Cancer Patients"

_cancers, 2021, doi:10.3390/cancers13051119_

Round 1

Reviewer 1 Report

In the present manuscript the authors describe the characterization of CTCs by their mechanical deformation profiles obtained with an optical stretcher. The CTCs are enriched using a density gradient centrifugation and further depletion of CD45+ cells. The remaining CD45- cells from healthy donors and breast cancer patients were further characterized using the optical stretcher.

The main finding is that CD45-cells in breast cancer patients and healthy donors differ significantly, suggesting that this difference is related with the presence of CTCs in the patient group.

Minor issue:

Line 263: Data according to Appendix Table A1 may not be normally distributed. Thus the median and range may be more suitable.

Line 304: Was the presence of CTCs confirmed in all 14 pats by immunofluorescence staining? CTC number added to Tabel A1 would be intresting to know.

Author Response

Minor issue:

Line 263: Data according to Appendix Table A1 may not be normally distributed. Thus the median and range may be more suitable.

The reviewer is correct, that median would be the obvious choice here. The mean proportion of CD45 negative cells in relation to the CD45 positive population was 3.5 % in samples from breast cancer patients, and 0.25% in samples from healthy donors. We added median and range of the CTC proportions in Table A1. In breast cancer patients the median proportion of CD45 negative cells in relation to the CD45 positive population was 0.92% ranging from 0.05 to 31.58%. In healthy donors, the median CD45-/CD45+ ratio was 0.33 % ranging between 0.1 and 0.4 %. Although the numbers do not look quite that impressive now, the statement of increased non-hematopoietic cell count in patients’ blood still holds. Since our data set is rather small, we assume the variance is due to heterogeneity rather than outliers. In case of absent outliers, the mean value is known to be more precise compared to the median. Therefore, we prefer to keep the mean proportion throughout the manuscript body though.

Line 304: Was the presence of CTCs confirmed in all 14 pats by immunofluorescence staining? CTC number added to Tabel A1 would be intresting to know.

Thank you for this interesting question. In total, we analyzed 3641 non-hematopoietic cells derived from 12 patients with breast cancer. We analyzed all the cells that were available in the remaining cell suspension after CD45 depletion using phase contrast images. The optical stretcher can analyze up to 350 cells per hour, but usually runs at 50-100 cells per hour with non-optimal samples and we left the machine running for up to 10 hours to analyze as many cells as possible. Unfortunately, we did not perform immunofluorescence staining on every sample. We will consider this remark for future analysis though.

Reviewer 2 Report

The authors of the manuscript »The mechanical fingerprint of circulating non-hematopoietic cells in breast cancer patients« report on the results of a study in which they compared mechanical properties of circulating non-hematopoietic cells (that served as a surrogate for circulating tumor cells) with healthy peripheral blood mononuclear cells. By studying cells of peripheral blood of 14 breast cancer patients they analyzed different mechanical properties by optical stretching over periods of few second using different wavelengths of laser. They have found elliptic deformation and shape restoration to be crucial discriminative parameters. By using machine learning algorithm, they were able to discriminate circulating tumor cells from normal blood cells.

The topic of the study is very interesting as well as clinically promising, since the future of cancer diagnostics as well as its therapy will probably depend on the distinction of tumor cells from normal cells in the peripheral blood. With technological advances of microfluidic and other chip-devices new discoveries on mechanic properties of cancer cells could become very useful. The introduction section of the presented manuscript describes some of the major current problems associated with detection of tumor cells in the peripheral blood of patients very nicely. In the methods section the experiments are clearly and thoroughly described. The results are nicely presented. The discussion section is also very clear and straightforward. The conclusions are sound and fair. The references sufficient, relevant and updated.

Author Response

Thank you for this positive review. We appreciate it very much.

Reviewer 3 Report

The paper is original and interesting proposing a novel method to detect cancer circulating cells into the blood stream by optical stretching analysis. The experiment is technically well performed. However a clinical design of the study is lacking.

Due to the novelty of the approach some issue should be better clarified as follows:

  • Results. What is the behavior at optical stretching of noncancer epithelial mammary cells?
  • Results. There was any difference in optical stretching among different breast cancer subtypes?
  • Discussion: specificity 0.63 is not very high and likely not enough to difference cancer patients form unaffected patients in a double blind study. Comparison with performances obtained by other methods should be more deeply reported and commented.
  • Discussion- A mechanical hypothesis explaining why epithelial cancer cells behave differently form noncancer epithelial cells should be provided.
  • Discussion. Can the use os mechanical fingerprint be integrated with other biomarkers (e.g., antigen analyses9 to increase performances?
  • Title is misleading: ‘circulating nonhematopoietic cells’ should be changed into ‘circulating tumor cells’.

Author Response

  • Results. What is the behavior at optical stretching of noncancer epithelial mammary cells?
  • Thank you for taking interest in the background of our study. Here, we aimed at finding mechanical differences between PBMC and non-hematopoietic cells, which might be potential CTC candidates, resulting from well-established and repeatable CD45 depletion method of PBMC from the blood of cancer patients. However, we have included a passage about the mechanical behavior of healthy breast tissue cells in comparison to breast cancer cells in the discussion section. The physical properties of tumor versus normal cells were already described elsewhere as we cited using the following references:

Fritsch A, Höckel M, Kiessling T et al. (2010) Are biomechanical changes necessary for tumour progression? Nature Phys 6(10):730–732. doi: 10.1038/nphys1800

Remmerbach TW, Wottawah F, Dietrich J et al. (2009) Oral cancer diagnosis by mechanical phenotyping. Cancer research 69(5):1728–1732. doi: 10.1158/0008-5472.CAN-08-4073

Ward KA, Li WI, Zimmer S et al. (1991) Viscoelastic properties of transformed cells: role in tumor cell progression and metastasis formation. Biorheology 28(3-4):301–313. doi: 10.3233/bir-1991-283-419

  • Results. There was any difference in optical stretching among different breast cancer subtypes?

Thank you for this interesting aspect. It would certainly be interesting to investigate mechanical differences between CTC of patients with various molecular subtypes. Indeed, this is one of our next project steps. In this manuscript, however, we focused on the feasibility of the OS to distinguish CTC candidates from blood cells. The sample size was too small to draw conclusions among subtypes.

  • Discussion: specificity 0.63 is not very high and likely not enough to difference cancer patients form unaffected patients in a double blind study. Comparison with performances obtained by other methods should be more deeply reported and commented.
  • We did not intend to awake high expectations towards a novel diagnostic high throughput method. Our aim was not to present a fully developed isolation assay, but to present the approach and explore into the field of rare cell identification by mechanical fingerprints. It is certainly of interest how far machine learning can improve clinical decisions. This was, however, not the topic. We can only refer to the measuring data obtained in our research lab and subsequent mechanical analysis using machine learning. We are far away from clinical usage. There is an interesting publication (https://doi.org/10.1016/j.artmed.2020.101854) by Kakileti et al though, which describes the usage of machine learning algorithms over medically interpretable parameters that describes the metabolic activity inside the breast tissue and indicate the presence of a possible malignancy even in asymptomatic women. Perhaps one day we can extend our study to develop a risk score based on mechanical parameters in the blood using machine learning. Assuming our proof-of-principle leads to practical applications in the clinic, RF could provide a probability for how likely a patient has CTC candidates in the blood.
  • Discussion- A mechanical hypothesis explaining why epithelial cancer cells behave differently form noncancer epithelial cells should be provided. 
  • The reviewer is right, that we should provide information on why we expect carcinoma cells to change their mechanical makeup. During cancer progression, the cytoskeleton is heavily altered and cell mechanical change happens almost universally. We added a short paragraph, pointing to different sources that confirm cytoskeletal change.
  • Discussion. Can the use os mechanical fingerprint be integrated with other biomarkers (e.g., antigen analyses9 to increase performances?
  • Thank you for this good idea. The mechanical features can certainly be combined with other parameters. We are planning on a combined analysis with hormone receptors and cell surface receptors such as HER2 and EpCAM. We added more emphasis on this part of the discussion.
  • Title is misleading: ‘circulating nonhematopoietic cells’ should be changed into ‘circulating tumor cells’.
  • We gladly changed the title of the manuscript.

Reviewer 4 Report

cancers-1102334 Review

The mechanical fingerprint of circulating non-hematopoietic cells in breast cancer patients

This article is written well, but lacks in new knowledge and grounds. Therefore, I require Major Revision.

・Please clarify the purpose of this study in “Introduction” section.

・Please clearly define the definition of circulating tumor cells (CTCs).

・Please make a paper with logically necessary data, not the amount of data.

・This is an important point. The number of cases in this study is far too small. You should increase the number of cases and conduct study again.

・Please validate from breast cancer intrinsic subtype (molecular).

・Also, luminal A and luminal B should be separated.

・In the analysis, please list the report by the statistician.

・And, please describe the discussion in more detail.

Minor point

・The sentence of this paper has many careful mention errors. Please review it.

Author Response

We thank the reviewer for the constructive criticism. The novelty of our approach is based on the mechanical aspect of cell biology. For the first time, we investigated the cell deformation behavior concerning the active and passive mechanical resistance of CD45 negative cells from the blood of breast cancer and present our findings as a proof of premise of mechanical phenotyping. We were able to show that CTCs exhibit mechanical features that might favor their escape from the vascular system.

・Please clarify the purpose of this study in “Introduction” section.

There is an enormous variety regarding CTC enrichment techniques available at the time being. However, active and passive mechanical features of CTC subtypes are not commonly part of CTC analysis. Due to phenotypical, mechanical, and functional changes during cancer progression as well as treatment, isolation of CTC subpopulations remains challenging and cannot be guided by the phenotype alone; cell mechanics have to be considered equally. Therefore, we investigated whether cell morphology together with mechanical deformation patterns might be an appropriate tool for marker-free CTC detection in the peripheral blood of patients with breast cancer. We believe that the promising combination of OS and random forest analyses might be enhanced with diagnostic cellular or molecular markers to potentially improve future treatment decisions. We added a sentence in the introduction section concerning the purpose of this study.

・Please clearly define the definition of circulating tumor cells (CTCs).

We apologize for being neglective concerning the CTC definition in line 53 “Detected cells are defined as CTCs when they are positive for the surface marker EpCAM and the intracellular cytokeratins 8, 18 and 19. Furthermore, CTCs are supposed to have a negative counterstain against the leucocyte marker CD45 and a positive nuclear DNA staining.” Since our manuscript combines knowledge from various scientific backgrounds we happened to overlook certain basic aspects. We added a phrase in the introduction: One reason for metastatic relapse in patients with breast cancer is hematogenous spread during early disease stages when single tumor cells detach from the primary tumor site and enter the vascular system [1]. Therefore, circulating tumor cells (CTCs) in the blood might be useful as predictive markers […]

・Please make a paper with logically necessary data, not the amount of data.

Thank you for this constructive comment. The data presented in the manuscript was structured as the project was built up. We used healthy peripheral blood mononuclear cells (PBMC), human MDA-MB 231 breast cancer cells to create a CTC model system for proof of premise. For translational experiments, we isolated CTC candidates (CD45 negative cells) from blood samples of breast cancer patients. Cells were mechanically characterized in the optical stretcher (OS). Active and passive cell mechanical data were related with physiological descriptors by a random forest (RF) classifier to identify cell type specific properties. The overlap of tumor biology and cell mechanics required thinking in many directions and coverage of numerous scientific aspects. We apologize if the manuscript caused confusion. However, we feel that all data shown is necessary in order to make the results scientifically sound.

・This is an important point. The number of cases in this study is far too small. You should increase the number of cases and conduct study again.

This is indeed an important point. We would like to emphasize that this manuscript describes a proof op premise. We fully agree that the sample size is very small and the results might appear somewhat preliminary. However, it is still a feasibility study and we were able to show distinct mechanical differences between blood cells and CTCs.

・Please validate from breast cancer intrinsic subtype (molecular). Also, luminal A and luminal B should be separated.

This is certainly a very interesting comment. Indeed, the investigation of mechanical differences between CTCs of patients with various molecular subtypes is one of our next project steps. In this manuscript, however, we focused on the feasibility of the OS to distinguish CTC candidates from blood cells. The sample size was way too small to draw conclusions among subtypes.

・In the analysis, please list the report by the statistician.

All statistical analyses presented in the manuscript were simply two-sample Kolmogorow-Smirnow-Tests between two data sets, as distributions were typically not Normal. They were performed by the authors themselves using Matlab 2019b. We did miss to include this information and would like to thank the reviewer for catching this detail.

・And, please describe the discussion in more detail.

We apologize if we did not appropriately elaborate our argumentation. We went over the discussion section again and extended points we deemed poorly explained.

Minor point

・The sentence of this paper has many careful mention errors. Please review it.

            Done.

Round 2

Reviewer 3 Report

All comments have been properly addressed.